# Multi-mode excitation drives disorder during the ultrafast melting of a C4-symmetry-broken phase

Daniel Perez-Salinas[1,4], Allan S. Johnson [1,4], Dharmalingam Prabhakaran [2] & Simon Wall [1,3✉]

Spontaneous $C_4$-symmetry breaking phases are ubiquitous in layered quantum materials, and often compete with other phases such as superconductivity. Preferential suppression of the symmetry broken phases by light has been used to explain non-equilibrium light induced superconductivity, metallicity, and the creation of metastable states. Key to understanding how these phases emerge is understanding how $C_4$ symmetry is restored. A leading approach is based on time-dependent Ginzburg-Landau theory, which explains the coherence response seen in many systems. However, we show that, for the case of the single layered manganite $La_{0.5}Sr_{1.5}MnO_4$, the theory fails. Instead, we find an ultrafast inhomogeneous disordering transition in which the mean-field order parameter no longer reflects the atomic-scale state of the system. Our results suggest that disorder may be common to light-induced phase transitions, and methods beyond the mean-field are necessary for understanding and manipulating photoinduced phases.

[1] ICFO – The Institute of Photonics Sciences, The Barcelona Institute of Science and Technology, 08860 Castelldefels, Barcelona, Spain. [2] Department of Physics, Clarendon Laboratory, University of Oxford, Oxford OX1 3PU, UK. [3] Department of Physics and Astronomy, Aarhus University, Ny Munkegade 120, 8000 Aarhus C, Denmark. [4] These authors contributed equally: Daniel Perez-Salinas, Allan S. Johnson. ✉email: simon.wall@phys.au.dk

As new light-induced hidden phases are discovered with properties like anomalous metallicity[1–3] and superconductivity[4,5], understanding how light can manipulate quantum materials is increasingly important[6]. Materials that show an instability between a $C_4$ symmetry-broken phase and a metallic or superconducting state in equilibrium appear especially likely to host such phases[7], and it is argued that photoexcitation suppresses the symmetry-broken phase[8], enabling the "hidden phase" to emerge before thermalization. For example, light-induced superconducting states have been attributed to preferential melting of stripe[4,5] or charge density wave[9] order, while insulator-metal phase transitions have been explained by selective melting of charge and orbital order[1–3]. However, the process by which the symmetry is restored following photoexcitation, and how energy couples to the different degrees of freedom, remains controversial and is critical for understand the microscopic origin of emergent transient and metastable phases in these materials.

The difficulty arises because solids have many degrees of freedom making it hard to track them all on the ultrafast timescale, particularly the order parameter (OP). Currently, the prevailing picture is a mean-field description, time-dependent Ginzburg-Landau (TDGL) theory, where the system is coarse-grained and partitioned into the OP and a potential determined by the other degrees of freedom. It is assumed that light transiently perturbs this potential, generating a force on the OP, and its subsequent evolution captures the key observed dynamics. TDGL makes several clear predictions when damping is minimal. Photoexcitation shifts the minima for the OP towards zero. If this shift is prompt, and small, the OP coherently oscillates about a new and reduced value[10]. The potential minimum evolves continuously with excitation until it reaches zero, while simultaneously softening, leading to critical slowing down of the OP dynamics[11]. At higher fluences, the potential minimum is fixed at the high-symmetry point, but the potential can still change and the OP can evolve coherently such that the system can overshoot and cross-over from one ordered domain to another, with an OP of opposite sign[10,12,13] and oscillates about the high-symmetry state. Such underdamped and coherent systems are amenable to coherent control[14], and this formulation has been applied to a range of systems, including the manganites[15], suggesting a universal behaviour in quantum materials and a way to realize non-thermal states.

While the vast majority of ultrafast phase transitions measured to date have been interpreted in this coherent framework[10–16], incoherent transitions have also been reported. The ultrafast phase transition in $VO_2$ was shown to result from broad distribution of modes, rather than a single OP for the entire system[17]. This results in rapid increase in atomic disorder as each unit cell undergoes a unique pathway to the high-symmetry state. As a result, the OP cannot transiently cross over to other domains during the phase transition, and once the order is lost, it remains lost. Therefore, determining which systems show disorder transitions and how the transient properties emerge is crucial for understanding transient phases in correlated materials.

It remains an open question whether TDGL can describe such order-disorder transitions. In principle, the individual trajectories of the underlying atomic system result in dephasing on a coarse scale, which can be incorporated in the TDGL as an overdamped response. In this case, the OP maps the coarse-grained mean response of the system. However, the mean dynamic does not have to act as a reliable measure for the local dynamics in the system. TDGL in particular, and mean-field theories more generally, assume that the local distribution of the OPs is compact around the mean value[18], even when spatial inhomogeneity is included[19,20]. This assumption can break down if the transition is heterogeneous, which is typical in first-order phase transitions. Figure 1 shows a spatially resolved OP for two pump-induced order-disorder scenarios where the mean OP is reduced by 50%. In one case, the system follows TDGL theory and the OP can take any value, even down to the atomic scale, and thus the distribution of local OP values is centred at 50%. Alternatively, a system could completely switch locally in 50% of the sample. Here the distribution is bi-modal, and no part of the system is actually in the state suggested by the mean. This distinction is particularly important because, if materials with first-order transitions in equilibrium exhibit dynamics governed by second-order-like TDGL, the OP can be manipulated non-thermally, whereas if first-order transitions cannot be described by TDGL, new theory beyond mean-field is needed to truly understand the transient properties of these materials.

In this paper, we present direct measurements of the OP dynamics in the $C_4$-symmetry restoring phase transition of the prototypical quasi-2D single layer manganite $La_{0.5}Sr_{1.5}MnO_4$ (LSMO) following ultrafast photoexcitation through ultrafast optical anisotropy. We chose this compound because the equilibrium properties have been extensively studied and the phase

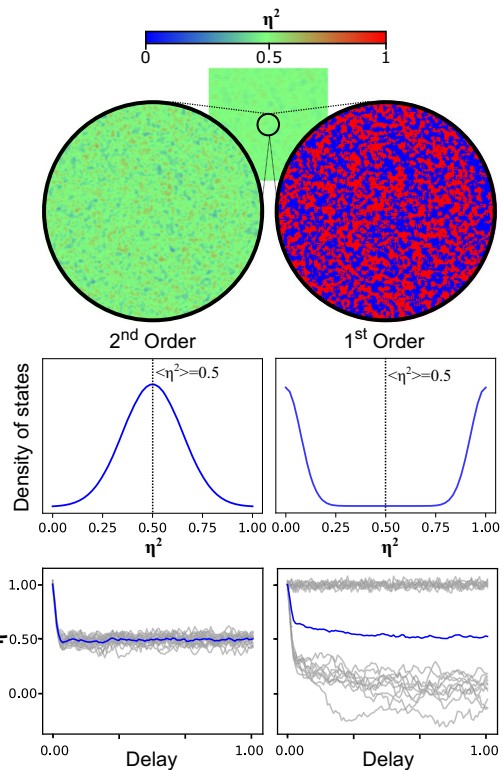

**Fig. 1 Beyond Ginzburg-Landau for atomic-scale dynamics.** A schematic of a spatially resolved order parameter measured on the mesoscale after photoexcitation during an ultrafast disordering transition for a second-order transition (left) and first order (right). An average value of $\eta^2 = 0.5$ is measured for the order parameter in both cases. For second-order transitions, the order parameter can take a continuum of values, and transitioning from the mesoscale to the atomic scale preserves the order parameter. The system is uniformly partially melted with a local order parameter distribution peaked at 0.5. The local dynamics are similar to the average, despite the disorder. However, for first-order transitions, the order parameter can take one of two values (0, 1). In this case, the atomic-scale view shows a bi-modal distribution peaked at either zero or 1, and no part of the system is at the averaged value, nor are the dynamics of the average order parameter reflected in the averaged response. Such distributions require physics beyond the mean-field to describe.

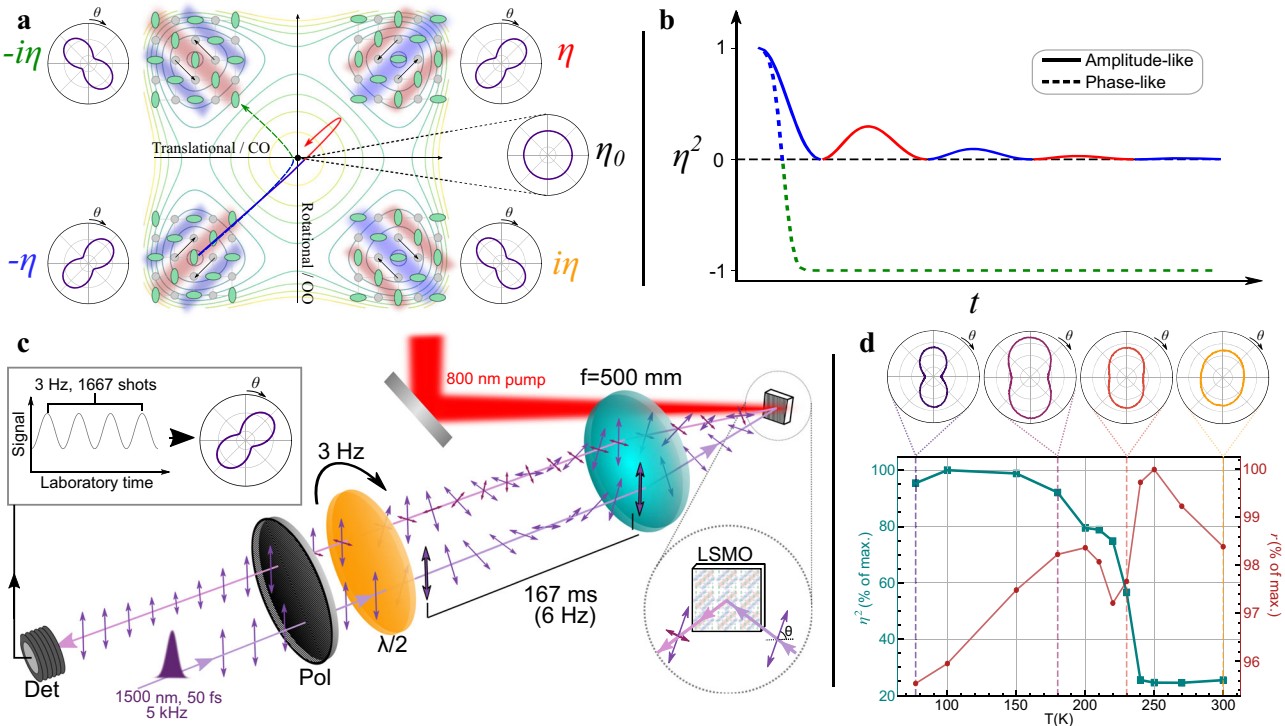

**Fig. 2 C4 symmetry breaking and reflection anisotropy in LaO.5Sr1.5MnO4. a** Schematic representation of the potential energy surface of the symmetry-broken phase. The four minima ($\pm\eta$, $\pm i\eta$) correspond to the directions along which the low temperature phase can form, relative to the high-symmetry structure, labelled $\eta_0$. Inserts show the reflection anisotropy signal for each domain structure, which can point along the [110] or [1–10] crystallographic axis. Domains of opposite parity have the same anisotropy and are thus indistinguishable. **b**. Expected signals for $\eta^2$ based on the two pathways shown in **a**. An amplitude-mode-like motion, in which the order is restored along a direct pathway, but overshoots, resulting in a "rebound" in the signal, and a phase-like motion, which flips the anisotropy axis (Line colour corresponds to the domain state). **c** Experimental setup for measurement of time-dependent anisotropy. A 60 fs 1500 nm probe pulse passes through a linear polarizer and rotating a halfwave plate. The probe is focused onto the LSMO sample, where the polarization component parallel to the input state is attenuated and a transverse component with a complex phase delay is introduced. The reflected beam is collected by the same lens and propagates back through the waveplate, where the initial polarization rotation is undone. The beam then passes back through the linear polarizer, removing the transverse polarization component introduced by the LSMO, and the modulated parallel component is collected on a photodiode. The real time-modulation of the signal is processed to yield the angular anisotropy signal (inset, top left). **d** $\eta^2$ and r as a function of temperature in our single-crystal LSMO sample. The polar plot radial axes run from 1.42 to 1.8 with divisions marked every 0.1 (arb. units). $\eta^2$ shows a clear discontinuity at $T_{CO}$ while r is less sensitive. The small non-zero value of $\eta^2$ above $T_{CO}$ is due to strain.

transition is first order[21,22]. Furthermore, the $C_4$ symmetry breaking dynamics in the related compound $Pr_{0.5}Ca_{1.5}MnO_4$ has been described in terms of TDGL theory, and manganites in general have been found to host a variety of emergent transient and metastable light-induced phases. We find that, while the OP dynamics qualitatively resemble those of an overdamped Ginzburg-Landau response, this similarity breaks down upon closer examination. Most strikingly, we find that the time scale for the phase transition decreases with fluence, and that a global change in the lattice potential occurs within 25 fs. The picture that emerges is one in which all accessible degrees of freedom are excited by the laser leading to rapid disorder and points towards physics beyond the mean-field approximation. Our results imply that all first-order phase transitions could respond in this way, but the process can be masked by initial state inhomogeneity. As inhomogeneity is common in first-order transitions, and is often invoked to explain pump-probe data[23,24], these results will impact a broad range of light-induced transitions.

## Results

LSMO undergoes a first-order insulator-to-semiconductor phase transition at $T_{CO} \approx 230\,K$, referred to as charge and orbital ordering. The phase transition involves the condensation of multiple phonon modes, primarily of oxygen character, which break the equivalence (charge ordering, CO) and isotropy (orbital

ordering, OO) of the Mn environment, changing the space group from $I4/mmm$ to $Cmnm$ and quadrupling the unit cell[21]. To date, most probes of the OP have focused on diffraction, however, the reduction to $C_2$ symmetry also results in electronic anisotropy and optical birefringence[8,25,26]. Two coefficients $r_1$ and $r_2$, the reflectivity parallel and perpendicular to the anisotropy axis, fully describe the *ab*-plane reflection anisotropy (RA). The anisotropy axis can lie along either the [110] or [1–10] directions of the crystal, resulting in multiple four possible domain configurations, as depicted in Fig. 2a. Just as with X-rays, optics can only distinguish two sets because the anisotropy is only sensitive to the square of the OP (see Supplementary Information S3). Domains situated in opposite quadrants of the phase diagram are connected by a 180° rotation and lattice translation, and are thus optically identical, whereas domains in adjacent quadrants are related by 90º rotations and have opposite optical anisotropies. The normalized OP, $\eta^2 = 2\frac{r_1 - r_2}{r_1 + r_2}$, and the isotropic reflectivity $r = \frac{r_1 + r_2}{2}$ can thus be determined by measuring the polarization-dependent reflectivity (see "Methods" for details).

Measuring the full RA pattern in the time domain can give a direct measurement of the OP dynamics following photoexcitation, in particular any coherent evolution. Figure 2b shows the evolution of $\eta^2$ expected for two such melting transitions. In the first case, an amplitude-mode-like transition (blue/red), the system moves directly to the high-symmetry D0 state but, due to its

momentum, continues to the opposite domain. The anisotropy drops to zero as the system passes through the D0 state and then "rebounds" as the system overshoots due to the indistinguishable nature of the domains. Similar rebounds in the OP are frequently claimed in the dynamics of X-ray diffraction peaks of many systems[10,12,14]. Alternatively, domain rotation could occur (blue/green). In this case, the system again shows an anisotropy decrease as the system reaches D0, before re-emerging with the opposite phase, i.e. the anisotropy signal switches sign.

To probe the dynamics we built a high-speed RA setup (Fig. 2c) that is compatible with ultrafast laser pulses. A high-speed spinning waveplate is used to modulate the polarization state of the laser shot-to-shot, enabling us to measure a full RA pattern in ~170 ms of acquisition and with 80 fs time resolution (see "Methods" for further details). The temperature dependence of $r$ and $\eta^2$ measured with our RA setup is shown in Fig. 2d. The isotropic part of the reflectivity is temperature dependent, but has no clear marker for $T_{CO}$, whereas the anisotropic component is small and constant above $T_{CO}$, and dramatically increases in magnitude as the sample is cooled below $T_{CO}$ and $C_4$ symmetry is broken.

In Fig. 3a, we show the restoration of $C_4$ symmetry in LSMO at 180 K after excitation with an 800 nm pump pulse. The RA pattern at several delays when the $C_4$ symmetry is partly (black) and completely (orange) recovered. The isotropic reflectivity notably increases while the OP decreases; see Supplementary Videos 1 and 2. At this temperature, high fluences can restore the $C_4$ symmetry without accumulative heating effects. At lower temperatures, high fluences could generate a metastable state, as found in other manganites[27–29], making the results there unreliable.

To get quantitative insight into the dynamics, we plot the time dependence of $\eta^2$ and $r$ for several fluences in Fig. 3b, c. At low fluences the isotropic reflectivity shows a fast increase and large amplitude coherent oscillations at 2.7 THz, followed by a slow exponential decay. However, at higher fluences the initial jump is suppressed and a slow rise is seen, together with a suppression of the phonon signal. These isotropic dynamics bear little resemblance to the response of the OP, which shows a rapid and incoherent suppression. As the fluence is increased, the OP suppressed to a greater degree and eventually saturates at the strain level observed above $T_{CO}$. While it is clear that a coherent TDGL theory cannot explain this data, an overdamped TDGL response may still be applicable. Indeed, the fluence dependence of the OP (Fig. 4a) shows a continuous decrease with fluence and no evidence of a threshold, consistent with the thresholdless response expected from TDGL. Therefore, intermediate values of the OP in LSMO could be interpreted as a transient non-thermal structure and indicate that the overdamped response of LSMO is fundamentally different from $VO_2$, despite both systems showing first-order transitions in equilibrium.

However, according to TDGL theory, the OP dynamics should slow down the closer the system gets to complete melting of the phase transition. In direct contrast, the normalized response of the OP shown in Fig. 4b shows that the OP actually speeds up as the system approaches the phase transition. We note that the reflectivity dynamics (Fig. 4c) do slow down when pumped across the phase transition, as has been interpreted as a signature of critical slowdown[11] or a bottle-neck timescale[2,26,30] in other materials, but here is found to be unrelated to the OP. Notably the isotropic reflectivity shows significant probe wavelength

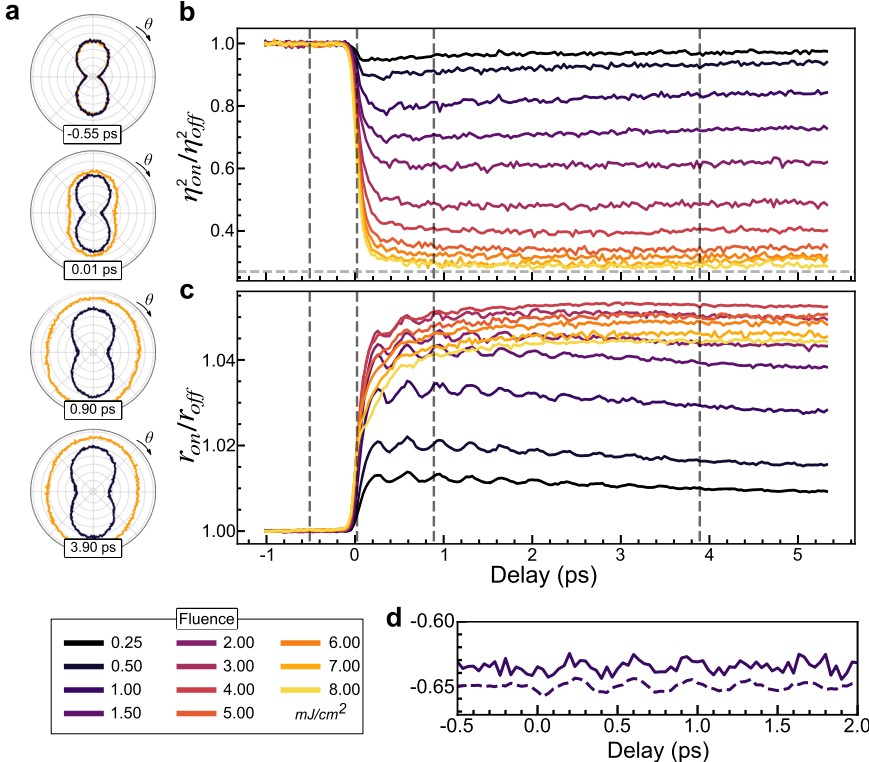

**Fig. 3 Time-dependent reflection anisotropy measurements of LSMO during the photoinduced phase transition. a** RA patterns as a function of pump-probe time delay for high (orange) and low (black) fluence. The radial axis runs from 1.63 to 2.015, with divisions every 0.05 (arb. units). **b** Ultrafast evolution of the normalized order parameter $\eta^2$ as a function of pump fluence which shows an ultrafast suppression. **c** Ultrafast evolution of the isotropic reflectivity $r$, showing a coherent phonon oscillations at 2.7 THz. **d** comparison of $\eta^2$ and $r$ at 0.5 mJ cm$^{-2}$ showing the 2.7 THz mode weakly modulates the RA signal. Supplementary Video 1 and Supplementary Video 2 show the RA pattern of each time delay for pump fluences of 1.5 and 8 mJ cm$^{-2}$, respectively, together with the corresponding plots of $r$ and $\eta^2$.

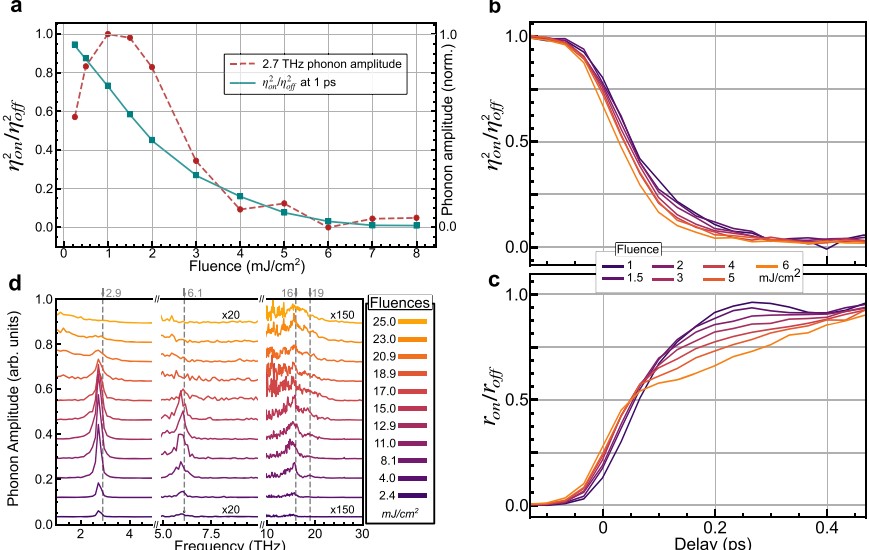

**Fig. 4 Simultaneous excitation of multiple degrees of freedom in LSMO. a** Fluence dependence of the order parameter and the amplitude of the 2.7 THz mode from r. The order parameter shows a monotonic decrease whereas the phonon amplitude first rises before being suppression at the same fluences as the order parameter. **b** Focused and normalized view of the first 500 fs of $\eta^2$ showing a marked speedup with increasing fluence. **c** Same as **b** but for r. In contrast to $\eta^2$ a slowdown is observed with increasing fluence. **d** Few-cycle pump-probe measurements of phonon dynamics with 1800 pump and 650 nm probe pulses (see methods and Supplementary Fig. S5 for details). 2.7, 6, 16 and 19 THz modes of LSMO are clearly observed and in good agreement with the literature[33] (dashed lines). The 2.7 and 6 THz modes are suppressed as the system is pumped through the transition, while the high-frequency modes broaden.

dependence, while the anisotropic dynamics do not (Supplementary Fig. S4), indicating the reflectivity measures dynamics of different degrees of freedom.

To better understand the transient state, we examine the response of the lattice through the coherent phonons. The 2.7 THz mode, which modulates the reflectivity, has a weak effect on the OP (Fig. 3d). It is known from Raman measurements that this mode is not the amplitude mode of the transition, but rather represents motion of the La/Sr ions[15,31] which play no role in the transition in equilibrium. The slight modulation seen then implies that either the phonon weakly drives the OP, or the mode itself generates an additional anisotropy independent of the OP, similar to strain. However, the concomitant loss in phonon signal with the suppression of the OP suggests an intimate connection between both degrees of freedom at the phase transition despite their weak coupling.

Due to the quadrupling of the unit cell during the phase transition, multiple phonon modes, that are otherwise forbidden in the high-symmetry state, become Raman active in the low-temperature phase and thus can potentially be excited by the laser. While the 2.7 THz mode does not play an active role in the phase transition, high-frequency Jahn–Teller modes have been suggested as being responsible for the transition[32]. To access these modes, we use sub-15 fs resolution transient reflectivity measurements (see "Methods"). Figure 4d shows that 2.7, 6, 16 and 19 THz modes can all be observed at low fluence, consistent with equilibrium Raman scattering data[33]. Importantly, all modes show changes when the OP is suppressed, with the 2.7 and 6 THz modes also becoming suppressed and the 16 and 19 THz modes broadened into a single peak. The concomitant changes suggest the entire potential for all degrees of freedom is perturbed, and not just a subset directly connected to the OP as assumed in TDGL. This change must be occurring faster than the highest frequency mode we measure, suggesting a sub-25 fs timescale. The excitation of a large number of modes then describes an order-disorder transition.

To understand how to interpret our data beyond TDGL, we investigate the role of inhomogeneity in the system. In first order phase transitions inhomogeneity is known to locally modify the critical temperature, and in LSMO in particular, it is known that the orbital ordering melts at the surface before the bulk[34]. As our probe is absorbed in less than the first 100 nm[15], we are sensitive to this effect and it, together with other sources of inhomogeneity, results in the smooth thermal transition experimentally observed in Fig. 2d. Importantly, the continued increase in orbital order below $T_{CO}$ is not due to an increasing magnitude of a local order, as would be the case for a homogeneous second-order transition, but results from more of the sample becoming ordered. We note that a coherent but inhomogeneous transition should still show coherent signatures in the OP[15], thus the order-disorder result is independent of this contribution.

This can have a major influence on understanding the fluence dependence, as shown in Fig. 5. Close to $T_{CO}$, pumping results in pre-melting of the orbital surface. Any energy absorbed by the laser can then move the ordered surface deeper into the bulk and there is no threshold to melt the surface of the system, even if a critical fluence dependence would be observed in a homogenous system. The system completely melts once the ordered surface has been pushed deep into the bulk and out of the probed volume (Fig. 5b). Thus, the thresholdless behaviour earlier taken as evidence of a second-order-like TDGL can be explained without invoking a different order for the phase transition in- and out-of-equilibrium. By repeating our measurements at 1200 nm, which penetrates less deeply than 1500 nm, we confirm the critical role of inhomogeneity in the system (Fig. 5d). While this fluence dependence can be described within TDGL theory, if the critical exponent is allowed to be different from the thermodynamic value, our approach does not need to consider such a change in behaviour (see Supplementary Information S7 for a more detailed discussion on fluence dependence within TDGL). A discrete, first-order-like switching also explains the lack of softening seen in the Raman modes as only the regions which are ordered oscillate, while those that are disordered cannot.

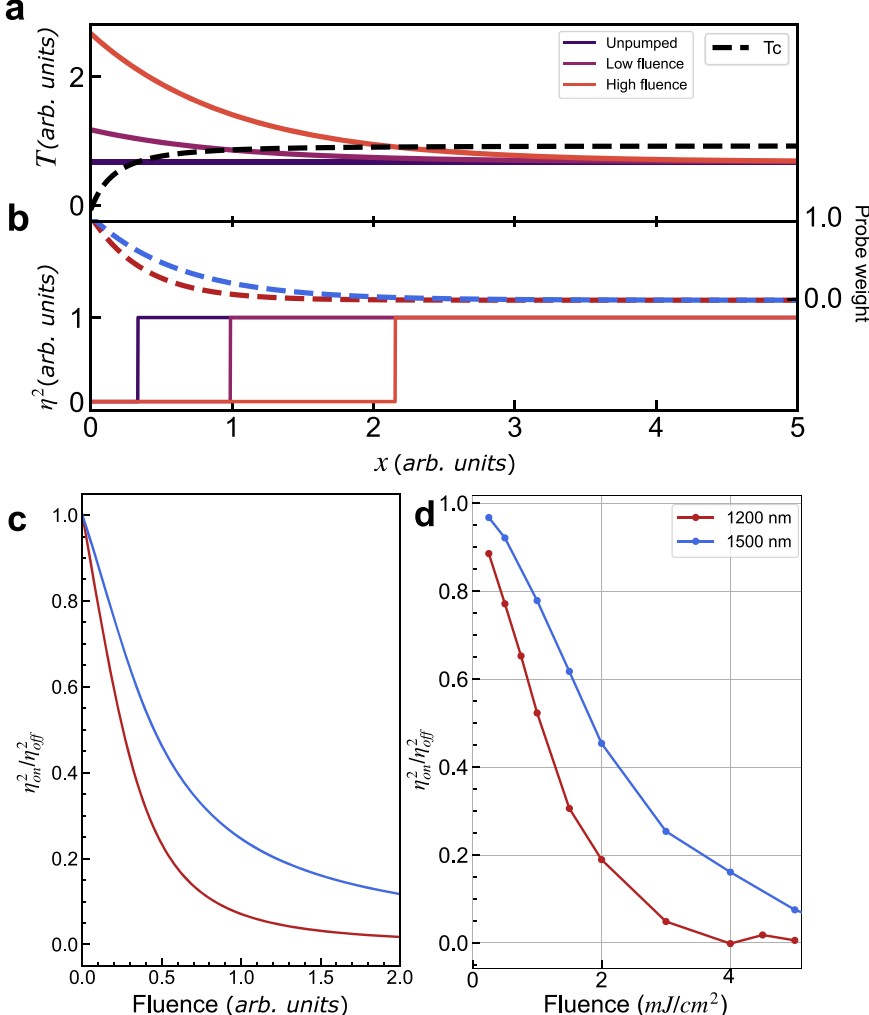

**Fig. 5 Inhomogeneity in the photo induced phase transition. a** First-order phase transitions can show surface melting, with the orbital of the phase transition melting at a lower transition temperature than the bulk. The black dashed line shows the thermodynamic transition temperature as a function of position in the sample from the surface at $x = 0$, together with two pump-induced temperature profiles. **b** The effect of the pump on the spatial dependence of the order parameter. Due to surface melting, part of the sample is already melted before the pump, resulting in a phase front. This phase front moves deeper into the bulk for harder excitation. The dashed lines indicate the weighting of the local order parameter on the averaged order parameter measured by probes which penetrate two different depths. **c** The measured order parameter as a function of pump fluence for two different probe conditions. Probes that penetrate less into the bulk reach saturation at a lower fluence than those that penetrate deeper. **d** Experimental data showing that changing the probe wavelength to 1200 nm reaches saturation before 1500 nm, demonstrating the role of spatial inhomogeneity in the probe.

## Discussion

Our results show that photoexcited LSMO undergoes an inhomogeneous and local ultrafast disordering, where all degrees of freedom are excited simultaneously and not just those required for the phase transition. The resulting dynamics then take place in a multi-dimensional phase space and when the system is more strongly excited, more phase space becomes accessible enabling the system to disorder faster. Such an effective change in dimensionality seems to be almost instantaneous, with the lattice responding to the change in symmetry in less than one-half period of the high-frequency mode (25 fs). This points to physics beyond TDGL, which is a low dimensional theory in which the system is necessarily coarse-grained and the main dynamics are confined to the OP in a restricted phase space of the global system. The close proximity to a metastable state in our sample also suggests a shared disorder origin for metastable states observed across the manganites[27,29].

While only $VO_2$ and LSMO have been described in terms of order–disorder transitions to date, the local dynamics which characterize first-order phase transitions in equilibrium naturally lend themselves to inhomogeneous, disordered transitions. In spite of this, many transient phase changes in first order materials, such as charge density wave systems, have been modelled with TDGL. It then remains an open question as to if and why some $C_4$-symmetric systems should a fundamentally non-thermal and coherent response while others show disorder. Key to resolving this issue will be to understand how inhomogeneity both before[35] and after excitation could impact interpretations based on TDGL theory. To this end, the need for techniques that can image the initial inhomogeneity as well as their dynamics[36,37] will be increasingly important[38–41]. Alternatively, disorder and fluctuations may stabilize the equilibrium high temperature phase in both $VO_2$[42] and LSMO[22] as both systems show large fluctuations above $T_c$, and systems that are stabilized by entropy may show fundamentally different dynamics to those driven by soft modes.

Disordered states in quantum materials in equilibrium are emerging as a new route to explain and generate new phenomena[43].

To date, explanations of effects such as light-induced superconductivity, have focused on possible non-equilibrium, ordered transient crystal structures[44] that would be expected from TDGL approaches. However, these systems also show large amounts of light-induced inhomogeneity and the restoration of $C_4$ symmetry would also be expected to generate disorder. Thus, our results hint that ultrafast disorder may in fact be responsible for the desirable properties in layered quantum materials.

## Methods

**Sample growth and characterization**. A single crystal of LSMO was grown using the optical float-zone method and cut with a 001 surface, then polished to an optical finish. All measurements presented here were performed on the same crystal. The transition was characterized using the temperature-dependent magnetization, which shows a single peak at $T_{CO}$ in agreement with the RA measurements. Samples from the same batch were characterized by X-ray diffraction for their charge and magnetic order peaks.

**Reflection anisotropy setup**. The RA was measured with linearly polarized 1500 nm pulses of 60 fs duration at 5 kHz repetition rate. A halfwave plate rotating at 3 Hz was used to probe the sample with all the possible linear polarization angles. For the time resolved experiment, 800 nm pump pulses of 50 fs duration at 5 kHz repetition rate were used. An important aspect of our setup is that the polarization state on the majority of optical elements is constant, enabling precise measurements of the RA signal. The LSMO sample was held in an open cycle nitrogen cryostat. Some inhomogeneity in the RA signal as a function of the position on the sample was observed, and so the probing position was kept constant by correcting any thermal dilation effects. More details can be found in Supplementary Information S1–S6.

**High time resolution setup**. High-frequency phonon dynamics were studied using 1800 nm pump pulses of 11 fs duration and 650 nm probe pulses of 10 fs duration at 1 kHz repetition rate. A time resolution of 25 fs, sufficient for observing the fastest phonon modes in LSMO, was determined from their cross correlation in a fifteen micron thick BBO crystal (Supplementary Fig. S2). Further details of these sources can be found in Johnson et al.[45] and Amuah et al.[46] An achromatic ultrafast spectroscopy system compatible with few-cycle duration pulses was used to perform the pump-probe measurements at near-normal incidence. The LSMO sample was held in a closed cycle helium cryostat at 180 K, and pump-induced changes to the probe reflectivity were recorded with a pair of diodes sampling the parallel and perpendicular polarization components. The traces in Fig. 4d average over polarization-dependent effects. The pump spot size was set to ≈50 times larger than the probe to ensure homogenous excitation, and the pump beam was chopped at 250 Hz.

**Fluence dependence model**. We use a simple 1D model of the LSMO OP in which the critical temperature varies according to a power law from the surface. Regions with a temperature higher than the critical temperature have OP = 0, while those below have OP = 1; the probe beam samples an exponentially decaying region of the sample according to the wavelength, and we integrate this sampling to return the effective OP observed. The critical temperature curve was adjusted to approximate the thermal first-order transition observed. Optical pumping was modelled as an exponentially decaying additive temperature, see Supplementary Information S7 for more information.

## Data availability

The authors declare that all data supporting the findings of this study are available within the paper and its supplementary information files. Source data are provided with this paper.

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

## Acknowledgements

We acknowledge insightful discussions with M. Eckstein. This work was funded through the European Research Council (ERC) under the European Union's Horizon 2020 Research and Innovation Programme (Grant Agreement No. 758461), by the Ministry of Science, Innovation and Universities (MCIU), State Research Agency (AEI) and European Regional Development Fund (FEDER) PGC2018-097027-B-I00, the Spanish State Research Agency through the "Severo Ochoa" programme for Centers of Excellence in R&D (CEX2019-000910-S), the Fundació Cellex, and Fundació Mir-Puig, the Generalitat de Catalunya through the CERCA programme, and EPSRC (grant no. EP/H033939/1). A.S.J. acknowledges support from Marie Skłodowska-Curie grant agreement No. 754510 (PROBIST).

## Author contributions

D.P. designed and built the RA setup and performed the RA measurements assisted by A.S.J. A.S.J. designed and built the high temporal resolution setup and performed measurements together with D.P. D.P. grew and characterized the sample. S.W. conceived of and organized the project. All authors participated in the writing of the manuscript.

## Competing interests

The authors declare no competing interests.
