## [Peer Review File · Nature Communications]

Reviewer #1 (Remarks to the Author):

With their rebuttal letter, the authors have given a detailed reply to my remaining points. While some of my questions have been convincingly answered, there are several points that I cannot completely agree to, as detailed below. I do not believe these should prevent publication in *Nature Communications*, however I would like the authors to consider these points and include some respective comments in the final manuscript.

Ad 2)

We do not believe that there is any evidence that the 2.7-3 THz mode in ref 9 actually relates the low and high temperature structures. In particular, this mode is known to involve motion of the A site ion, which is also known not to change during the transition, thus it seems hard to reconcile how this mode can really represent the transition.

A Raman active mode modulating a superlattice Bragg peak intensity does not mean that it is affecting the atoms that drive the transition. In LSMO there are 39 Raman active modes, of which 11 are Ag symmetric. All 11 of these modes can change the intensity of the superlattice peaks. However, in LSMO it is also known that only 3 structural modes drive the transition.

We also feel evidence for frequency doubling in Ref 9 is not strong. LSMO has fundamental modes at 2.9 and 6 THz in Raman scattering. These are almost the double of each other, but as they are observed in equilibrium Raman scattering, they cannot be interpreted as mode doubling.

In our opinion, there is no evidence to exclude the 6 THz mode observed in ref 9 as being a low symmetry Raman active mode. However, a detailed criticism of the work in ref 9 is beyond the scope of this paper/referee reply.

While it seem a bit aside the point to discuss a different manuscript here, I need to disagree with the statements of the authors here. Ref. 9 indeed shows that the positions of the A site ions change across the transition, and the sum of the modes considered there corresponds to the change between high and low temperature structures. I agree that principally all Raman active modes can modulate the SL intensities, but only modes that undo the symmetry changes between the structures will lead to complete suppression of SL reflections. This is clearly demonstrated for the ~3 THz mode in Ref. 9. The observed frequency doubling when this mode crosses the high-temperature position (with vanishing SL intensity) is also reproduced by the employed structural model, which is a very strong argument against the involvement of other modes, as suggested in the authors' reply.

Ad 3)

We disagree with the referee here and, while the critical exponent used will affect the quantitative aspects fluence dependence, it does not affect the observation of the kink. As described in the text the kink appears when the order parameter value reaches zero at the surface. The kink will always be observed in a homogeneous system because the surface is always ordered. Changing the critical exponent (p) only changes the contrast, see the figures below. Of course, observation of kink will then rely on appropriate signal to noise.

I agree that there should be a faint resemblance of a “kink” also for higher values of the critical exponent, however I’m not convinced that the data the authors present allows distinguishing these subtleties in the fluence dependence. Indeed, looking closely at Fig. 5d, one could argue to distinguish a faint “kink” in the data at a similar fluence:

I suggest the authors at least mention such an alternative explanation in the manuscript.

Reviewer #2 (Remarks to the Author):

The authors have convincingly answered the questions in my previous report. Based on that, and on my earlier reports, I recommend publication in Nature Communications.

I somewhat agree with the statement by referee 3, that the importance of disorder in photo-induced phase transitions may not be too surprising at a first order phase transition, in particular given the known importance of inhomogeneities for quantum materials even in equilibrium. It is also clear that the correct interpretation of the dynamics in this case this requires an approach beyond TDGL (or rather, beyond the Gaussian approximation within TDGL). Nevertheless, I would say that this fact is mostly not considered in interpreting the sub ps dynamics of photo-induced phase transitions, also because this would be very challenging from a theoretical perspective. For this reason, I do anticipate an important impact of the present study.

Reviewer #3 (Remarks to the Author):

I have read the paper and the comments, and I think the results are of sufficient impact to warrant publication in Nature Communications. In regards to the comment of the authors "Our point was that, to the best of our knowledge, the presence of inhomogeneity in the initial state has not been used to interpret data." There is quite a bit of literature (theory and experiment) where the dynamics depend on the inhomogeneous nature of the ground state. Here is a partial list on several quantum materials (I'm not saying these papers all need to be cited, but there is a lot of recent work in this direction).

1. Ultrafast switching to an insulating-like metastable state by amplitudon excitation of a charge density wave

<https://doi.org/10.1038/s41567-021-01267-3>

2. Subterahertz collective dynamics of polar vortices

<https://doi.org/10.1038/s41586-021-03342-4>

3. Inhomogeneity of the ultrafast insulator-to-metal transition dynamics of VO₂

<https://doi.org/10.1038/ncomms7849>

4. Ultrafast Nanoimaging of the Photoinduced Phase Transition Dynamics in VO₂

<https://doi.org/10.1021/acs.nanolett.5b05313>

5. Nanotextured Dynamics of a Light-Induced Phase Transition in VO₂

<https://doi.org/10.1021/acs.nanolett.1c02638>

6. Domain Dynamics under Ultrafast Electric-Field Pulses

<https://doi.org/10.1103/physrevlett.124.107601>

7. Role of equilibrium fluctuations in light-induced order

<https://arxiv.org/pdf/2110.00865>

8. Multi-messenger nanoprobe of hidden magnetism in a strained manganite

<https://www.nature.com/articles/s41563-019-0533-y>

9. Nucleation and Growth Bottleneck in the Conductivity Recovery Dynamics of Nickelate Ultrathin Films

<https://pubs.acs.org/doi/10.1021/acs.nanolett.0c02828>

10. Ultrafast manipulation of mirror domain walls in a charge density wave

<https://advances.sciencemag.org/content/4/10/eaau5501?intcmp=trendmd-adv>

11. Pump-induced motion of an interface between competing orders

<https://doi.org/10.1103/physrevb.101.224305>

12. Ultrafast nanoimaging of the order parameter in a structural phase transition

<https://doi.org/10.1126/science.abd2774>

13. Ultrafast optical excitation of magnetic skyrmions

<https://doi.org/10.1038/srep09552>

14. Dynamic conductivity scaling in photoexcited V₂O₃ thin films

<https://journals.aps.org/prb/abstract/10.1103/PhysRevB.92.085130>

REVIEWERS' COMMENTS

Reviewer #1 (Remarks to the Author):

With their rebuttal letter, the authors have given a detailed reply to my remaining points. While some of my questions have been convincingly answered, there are several points that I cannot completely agree to, as detailed below. I do not believe these should prevent publication in Nature Communications, however I would like the authors to consider these points and include some respective comments in the final manuscript.

We thank the reviewer for their positive response and acknowledge there are plenty of open questions that require further debate

While it seem a bit aside the point to discuss a different manuscript here, I need to disagree with the statements of the authors here. Ref. 9 indeed shows that the positions of the A site ions change across the transition, and the sum of the modes considered there corresponds to the change between high and low temperature structures. I agree that principally all Raman active modes can modulate the SL intensities, but only modes that undo the symmetry changes between the structures will lead to complete suppression of SL reflections. This is clearly demonstrated for the ~3 THz mode in Ref. 9. The observed frequency doubling when this mode crosses the high-temperature position (with vanishing SL intensity) is also reproduced by the employed structural model, which is a very strong argument against the involvement of other modes, as suggested in the authors' reply.

As the referee states, this is primarily a discussion of another manuscript. The aim was to show that alternative explanations can explain other data, but a full debate needs a different forum

I agree that there should be a faint resemblance of a "kink" also for higher values of the critical exponent, however I'm not convinced that the data the authors present allows distinguishing these subtleties in the fluence dependence. Indeed, looking closely at Fig. 5d, one could argue to distinguish a faint "kink" in the data at a similar fluence:

I suggest the authors at least mention such an alternative explanation in the manuscript.

We have added the following text to the discussion of the kink in the manuscript:

While this fluence dependence can be described within TDGL theory, if the critical exponent is allowed to be different from the thermodynamic value, our approach does not need to consider such a change in behaviour. (see Supplementary Discussion 7)

We have also added a discussion and a figure in the Supplementary Information which point out the need for high signal-to-noise to detect the kink in the case for a critical exponent close to 1.

Reviewer #2 (Remarks to the Author):

The authors have convincingly answered the questions in my previous report. Based on that, and on my earlier reports, I recommend publication in Nature Communications.

I somewhat agree with the statement by referee 3, that the importance of disorder in photo-induced phase transitions may not be too surprising at a first order phase transition, in particular given the known importance of inhomogeneities for quantum materials even in equilibrium. It is also clear that the correct interpretation of the dynamics in this case this requires an approach beyond TDGL (or rather, beyond the Gaussian approximation within TDGL). Nevertheless, I would say that this fact is mostly not considered in interpreting the sub ps dynamics of photo-induced phase transitions, also because this would be very challenging from a theoretical perspective. For this reason, I do anticipate an important impact of the present study.

We thank the reviewer again for their positive impressions.

Reviewer #3 (Remarks to the Author):

I have read the paper and the comments, and I think the results are of sufficient impact to warrant publication in Nature Communications. In regards to the comment of the authors “Our point was that, to the best of our knowledge, the presence of inhomogeneity in the initial state has not been used to interpret data.” There is quite a bit of literature (theory and experiment) where the dynamics depend on the inhomogeneous nature of the ground state. Here is a partial list on several quantum materials (I’m not saying these papers all need to be cited, but there is a lot of recent work in this direction).

1. Ultrafast switching to an insulating-like metastable state by amplitudon excitation of a charge density wave

<https://doi.org/10.1038/s41567-021-01267-3>

2. Subterahertz collective dynamics of polar vortices

<https://doi.org/10.1038/s41586-021-03342-4>

3. Inhomogeneity of the ultrafast insulator-to-metal transition dynamics of VO₂

<https://doi.org/10.1038/ncomms7849>

4. Ultrafast Nanoimaging of the Photoinduced Phase Transition Dynamics in VO₂

<https://doi.org/10.1021/acs.nanolett.5b05313>

5. Nanotextured Dynamics of a Light-Induced Phase Transition in VO₂

<https://doi.org/10.1021/acs.nanolett.1c02638>

6. Domain Dynamics under Ultrafast Electric-Field Pulses

<https://doi.org/10.1103/physrevlett.124.107601>

7. Role of equilibrium fluctuations in light-induced order
<https://arxiv.org/pdf/2110.00865>
8. Multi-messenger nanoprobe of hidden magnetism in a strained manganite
<https://www.nature.com/articles/s41563-019-0533-y>
9. Nucleation and Growth Bottleneck in the Conductivity Recovery Dynamics of Nickelate Ultrathin Films
<https://pubs.acs.org/doi/10.1021/acs.nanolett.0c02828>
10. Ultrafast manipulation of mirror domain walls in a charge density wave
<https://advances.sciencemag.org/content/4/10/eaau5501?intcmp=trendmd-adv>
11. Pump-induced motion of an interface between competing orders
<https://doi.org/10.1103/physrevb.101.224305>
12. Ultrafast nanoimaging of the order parameter in a structural phase transition
<https://doi.org/10.1126/science.abd2774>
13. Ultrafast optical excitation of magnetic skyrmions
<https://doi.org/10.1038/srep09552>
14. Dynamic conductivity scaling in photoexcited V2O3 thin films
<https://journals.aps.org/prb/abstract/10.1103/PhysRevB.92.085130>

We thank the Reviewer for their positive impressions and the detailed literature list. We have now included several of these recommendations in the main text of the manuscript:

As inhomogeneity is common in first-order transitions, and is often invoked to explain pump probe data^{23,24}, these results will impact a broad range of light induced transitions.

Key to resolving this issue will be to understand how inhomogeneity both before³⁵ and after excitation could impact interpretations based on TDGL theory. To this end, the need for techniques that can image the initial inhomogeneity as well as their dynamics^{36,37} will be increasingly important³⁸⁻⁴¹.